# Trioxolone Methyl, a Novel Cyano Enone-Bearing 18βH-Glycyrrhetinic Acid Derivative, Ameliorates Dextran Sulphate Sodium-Induced Colitis in Mice

**DOI:** 10.3390/molecules25102406

**Published:** 2020-05-21

**Authors:** Andrey V. Markov, Aleksandra V. Sen’kova, Oksana V. Salomatina, Evgeniya B. Logashenko, Dina V. Korchagina, Nariman F. Salakhutdinov, Marina A. Zenkova

**Affiliations:** 1Institute of Chemical Biology and Fundamental Medicine, Siberian Branch of the Russian Academy of Sciences, Lavrent’ev ave., 8, 630090 Novosibirsk, Russia; alsenko@mail.ru (A.V.S.); ana@nioch.nsc.ru (O.V.S.); evg_log@niboch.nsc.ru (E.B.L.); marzen@niboch.nsc.ru (M.A.Z.); 2N.N. Vorozhtsov Novosibirsk Institute of Organic Chemistry, Siberian Branch of the Russian Academy of Sciences, Lavrent’ev ave., 9, 630090 Novosibirsk, Russia; korchaga@nioch.nsc.ru (D.V.K.); anvar@nioch.nsc.ru (N.F.S.)

**Keywords:** cyano enone, soloxolone methyl, 18βH-glycyrrhetinic acid, CDDO-Me, colitis, inflammation, molecular docking, molecular targets

## Abstract

Semi-synthetic triterpenoids, bearing cyano enone functionality in ring A, are considered to be novel promising therapeutic agents with complex inhibitory effects on tissue damage, inflammation and tumor growth. Previously, we showed that the cyano enone-containing 18βH-glycyrrhetinic acid derivative soloxolone methyl (SM) effectively suppressed the inflammatory response of macrophages in vitro and the development of influenza A-induced pneumonia and phlogogen-stimulated paw edema in vivo. In this work, we reported the synthesis of a novel 18βH-glycyrrhetinic acid derivative trioxolone methyl (TM), bearing a 2-cyano-3-oxo-1(2)-en moiety in ring A and a 12,19-dioxo-9(11),13(18)-dien moiety in rings C, D, and E. TM exhibited a high inhibitory effect on nitric oxide (II) production by lipopolysaccharide-stimulated J774 macrophages in vitro and dextran sulfate sodium (DSS)-induced colitis in mice, displaying higher anti-inflammatory activity in comparison with SM. TM effectively suppressed the DSS-induced epithelial damage and inflammatory infiltration of colon tissue, the hyperproduction of colonic neutral mucin and TNFα and increased glutathione synthesis. Our in silico analysis showed that Akt1, STAT3 and dopamine receptor D2 can be considered as mediators of the anti-colitic activity of TM. Our findings provided valuable information for a better understanding of the anti-inflammatory activity of cyano enone-bearing triterpenoids and revealed TM as a promising anti-inflammatory candidate.

## 1. Introduction

Michael acceptor groups are important structural elements of a broad range of pharmacologically active compounds, as they determine the direct interaction with biologically relevant targets. Cysteine thiol groups of proteins are considered one of the most susceptible sites to Michael acceptor additions, and the formation of thiol adducts with proteins underpins a high biological activity and a multi-targeted mode of action of Michael acceptor-bearing agents [1].

One of the most studied Michael acceptor pharmacophore groups is the α-cyano α,β-unsaturated carbonyl moiety. The modification of natural materials by this group was shown to significantly improve the native biological activity of the starting compounds [2,3]. The most promising results in this field were obtained for 2-cyano-3-oxo-1(2)-en bearing semi-synthetic derivatives of pentacyclic triterpenoids, including oleanane, ursane, lupane and the lanostane series. (Figure 1, compounds **A**–**C**) [4,5,6,7,8].

The formation of an additional Michael acceptor center at C-9 by creating the 12-oxo-9(11)-en moiety in the C ring of the triterpenoid scaffold along with the 2-cyano-3-oxo-1(2)-en system significantly enhanced the bioactivity of semi-synthetic triterpenoids (Figure 1) [9,10,11,12]. According to Liby and Sporn, the most promising results in this field were obtained for bardoxolone methyl (also known as CDDO-Me) (Figure 1), an oleanolic acid derivative that displays a wide spectrum of bioactivities, including marked antitumor, anti-inflammatory and cytoprotective effects in both cellular and animal models [4], reaching Phase III clinical trials for chronic kidney disease in diabetic patients [13], pulmonary hypertension [13] and Alport syndrome [14].

Previously, our research group synthesized and evaluated the bioactivities of soloxolone methyl (SM), a position isomer of bardoxolone methyl based on a 18βH-glycyrrhetinic acid scaffold [10]. We showed that SM effectively inhibited tumor cell growth in vitro [10] and in vivo [15] and displayed marked anti-inflammatory and anti-influenza A activities [15,16,17]. In our previous report, structure-activity relationship (SAR) analysis revealed that the acceptor center at C-9 in ring C was crucial for the bioactivity of SM—an analog of SM with a saturated 9(11)-double bond was found to display a significantly lower cytotoxicity and a significantly lower inhibitory effect on nitric oxide (II) (NO) production by inflamed macrophages compared to SM [16]. Thus, manipulations on the Michael acceptor site in ring C can be considered a promising approach to modulate the biological activity of cyano enone-bearing triterpenoids.

Here, we reported the synthesis and evaluation of the biological activities of a first generation derivative of SM: trioxolone methyl (TM), containing a novel 12,19-dioxo-9(11),12(13)-dien system in rings C, D and E along with the well known 2-cyan-3-oxo-1(2)-en moiety in ring A (Figure 1). We proposed that the introduction of an additional enone group in rings D/E could increase the bioactivity of the derivative due to an enhancing effect on the acceptor properties of the electrophilic site at C-9 in ring C and the expansion of the hydrogen bond network between triterpenoid and its protein targets. Previously, the introduction of an additional carbonyl group in ring E markedly reinforced the cytotoxicity and α-glucosidase inhibitory activity of lupane and corosolic acid derivatives, respectively [18,19].

Given the revealed ability of SM to suppress the inflammatory response of lipopolysaccharide (LPS)-stimulated J774 macrophages in vitro [16] and carrageenan-induced acute paw edema and influenza A-associated pneumonia in murine models [15,17], we questioned whether the formation of an additional enone moiety in rings D/E along with the enone system in ring C increased the anti-inflammatory properties of the molecule in vitro and in vivo. Here, we showed that TM effectively suppressed NO production by LPS-challenged J774 macrophages in vitro and dextran sulfate sodium (DSS)-induced colitis in mice in vivo. We found that the anti-inflammatory activity of TM was more pronounced compared to SM. Chemo- and bioinformatic approaches revealed Akt, STAT3 and dopamine receptor D2 as the probable primary targets of TM, associated with the anti-colitic activity.

## 2. Results and Discussion

### 2.1. Synthesis of TM

TM [methyl 2-cyano-3,12,19-trioxo-olean-1(2),9(11),13(18)-trien-30-oate] was synthesized using 2-cyano-3-oxo-18βH-olean-1(2),11(12)-dien-30-oate **1** as a starting material [16] (Scheme 1). The 12(13)-double bond in the C ring of compound **1** was converted into a 12,19-dioxo-9(11),13(18)-dien moiety by the treating of **1** with selenium dioxide in acetic acid under reflux. This transformation is well known for the β-amirin pentacyclic triterpenoids, notably oleanolic and 11-deoxoglycyrrhetinic acids [20]. Interestingly, the cyano enone fragment in ring A was inert to the action of the used oxidizing agent, as we did not observe the presence of any by-products associated with its destruction in the reaction mixture (the reaction course was monitored by ^1^H-NMR). The structure of TM was further confirmed by ^1^H and ^13^C-NMR and high-resolution mass spectrometry.

In comparing the ^1^H-NMR spectra for the compounds TM and SM [10], we observed a slight downfield shift of H-11 (6.08 ppm for TM and 5.97 for SM). The signals of CH_3_-24 and CH_3_-25 kept their place at ~1.14 ppm and ~1.47 ppm, respectively; and signals CH_3_-27, CH_3_-28 and CH_3_-29 of TM showed the expected downfield shifts (CH_3_-27: 1.13 ppm for TM and 0.96 ppm for SM; CH_3_-28: 1.03 ppm for TM and 0.90 ppm for SM; CH_3_-29: 1.57 ppm for TM and 1.09 ppm for SM). However, the signals CH_3_-23 and CH_3_-26 showed upfield shifts (CH_3_-23: 1.19 ppm for TM and 1.22 ppm for SM; CH_3_-26: 1.34 ppm for TM and 1.44 for SM), which may indicate a change in the shape of the TM molecule compared to SM. In the ^13^C-NMR spectrum, the 12,19-dioxo-9(11),13(18)-dien system was clearly represented: the additional signals of a double bond carbon at 138.6 ppm (C-13) and 152.5 (C-18) as well as a carbonyl atom at 205.9 ppm (C-19) were observed. The signal C-12 also showed upfield shifts (186.5 ppm for TM and 199.5 ppm for SM).

### 2.2. TM Inhibits NO Production by LPS-Stimulated J774 Macrophages

In order to assess the anti-inflammatory potential of TM in vitro, we analyzed the effects on the production of the pro-inflammatory mediator NO using murine J774 macrophages, challenged by a bacterial endotoxin, and compared this with the effect of SM. First, a 3-(4, 5-dimethylthiazol-2-yl)-2, 5-diphenyltetrazolium bromide (MTT) assay was performed to identify the non-toxic doses of the triterpenoids. As seen in Figure 2A, in concentrations up to 1 µM, both TM and SM did not induce cytotoxicity in J774 cells.

Then, LPS-stimulated J774 macrophages were treated with the tested triterpenoids and the nitrite concentration was measured in a culture medium using the Griess reaction. As can be seen from Figure 2B, both compounds effectively suppressed the synthesis of NO by inflamed J774 cells. The bioactivity of TM was more pronounced than that of SM: TM two-fold inhibited the NO production at 50 nM and declined it to a basal level at 1 µM, whereas SM at 50 nM did not affect this process, and its maximum used concentration (1 µM) suppressed the NO generation only by 2.8-fold in comparison with the control (Figure 2B). The identified ability of TM to hamper the inflammatory response of the macrophages induced by bacterial LPS in vitro demonstrated the expediency of a further more detailed investigation of its anti-inflammatory activity in animal models.

### 2.3. TM Inhibits DSS-Induced Colitis in Mice

To date, a number of studies have found that both natural pentacyclic triterpenoids and their derivatives can effectively ameliorate experimental colitis in rodent models [21,22,23,24,25]. Based on these findings, we questioned whether TM protected against dextran sulfate sodium (DSS)-induced colitis in mice and whether the efficiency of TM was higher than that of SM in this model. To understand this, acute colitis was induced in C57Bl/6 mice by the administration of 2.5% DSS in drinking water for seven days. The investigated triterpenoids were administered at a dosage of 5 mg/kg in sesame oil via gastric gavage during colitis induction and following three days recovery (Figure 3A); sulfasalazine (SLZ), administered similarly, was used as a reference drug.

Colon shortening is known to be tightly associated with colitis, and colon length is often used as a valuable parameter for the degree of inflammation [26]. As shown in Figure 3B, DSS causes significant changes in the colon state: the colon lengths of DSS administered to untreated and vehicle-treated mice were 1.3-fold shorter than those of healthy mice (Figure 3B). The administration of TM restored the colon length almost to the size of the healthy control; SM and SLZ also suppressed colon shortening to an extent; however, the differences between SM/SLZ-treated and vehicle-treated mice were statistically insignificant (Figure 3B).

The measurements of the body weight of the mice revealed weight loss in all DSS-treated groups, which was approximately 10–13% of the initial body weight on day 10 of the colitis induction (Figure 3C). The administration of TM displayed a protective effect during the first five days of DSS feeding; however, during the second half of the experiment, this effect was lost (Figure 3C). The administration of SM or SLZ were found not to attenuate DSS-induced body weight changes throughout the study, which was consistent with the absence of their effect on DSS-induced colon length shortening as mentioned above (Figure 3B).

Surprisingly, despite the revealed protective effects, TM did not effectively reduce the presence of blood in the stool, which is the one of the manifestations of colitis [27]; the calculation of the disease activity index (DAI) showed that TM caused improvement in the DAI only at days 7–8 of the study, whereas toward the end of the experiment, the DAI of the TM-treated mice was higher than that of the vehicle-treated group (Figure 3C). Interestingly, SM and SLZ showed more pronounced inhibitory effects on the presence of blood in the stool compared to TM: the DAI scores of SM- and SLZ-treated mice at day 10 were 3.4- and 7.1-times lower, respectively, compared to those of the vehicle-treated mice (Figure 3C).

Then, in order to evaluate the effects of semi-synthetic triterpenoids on the development of colitis more precisely and to investigate the ambiguous results obtained for TM (notably, the suppression of DSS-induced colon shortening along with the absence of a reduction of the DAI score), we performed a histological analysis of the colon tissue. As shown in Figure 4A, the colons of healthy mice showed intact epithelium, mucosa and submucosa, non-disrupted crypts, and goblet cells with mucus vacuoles. The administration of DSS caused severe colon tissue damage mainly confined to the distal colon and represented by a massive epithelium disruption with erosion and ulceration and the diffuse destruction of the crypt architecture (Figure 4A, DSS). A pronounced inflammatory infiltration of colons (represented predominately by neutrophils with a mixture of lymphocytes and macrophages) was also revealed, which in some cases was spread transmurally (Figure 4A).

The treatment of DSS-exposed mice with TM led to reductions in the epithelium disruption, crypt damage and inflammatory infiltration of colon tissue; the histological structure of the colons in this group was found to be comparable to that of healthy mice (Figure 4A). The cumulative histological score in TM-treated mice was 4.2- or 4.9-fold lower than that of untreated or oil-treated DSS-administered mice, respectively (Figure 4B). In the case of SM administration, the level of colon damage and inflammation in the colon tissue was comparable with that of the DSS-induced vehicle-treated group, whereas SLZ reduced the morphological manifestations of colitis; however, this effect was insignificant. The calculation of the cumulative histological scores revealed that these groups were not statistically different from the DSS-exposed untreated or vehicle-treated groups (Figure 4B).

Inflammation leads to reactive mucus hyperproduction, which represents a compensatory reaction to mucosal damage by inflammatory cells [28]. To examine the effect of the investigated triterpenoids on the neutral mucin production, we conducted the Periodic Acid-Schiff (PAS) staining of colon tissues. Plots for the PAS reactions were selected for those with severe inflammation without erosive and ulcerative epithelial lesions. The PAS staining revealed a significant increase in the level of the neutral mucin in all DSS-treated groups compared to the healthy control. The colon sections depicted in Figure 4C clearly demonstrate that TM and SLZ effectively inhibited the neutral mucin hyperproduction, which reflects their pronounced anti-inflammatory effects in DSS-inflamed colon tissues, whereas SM did not affect DSS-induced mucin synthesis.

Thus, the performed histological study and analysis of the mucin content in colon tissue clearly confirmed the anti-colitic activity of TM. Our obtained data indicated that the observed low efficiency of TM to reduce the level of visible blood in stool (Figure 4B) can be explained independently of its reparative effect on DSS-injured colon tissue. According to the ability of glycyrrhizin, a glycoside of glycyrrhetinic acid (a parent compound of SM and TM) to selectively inhibit thrombin, as reported previously by Mauricio et al. [29], we suggested that this peculiarity may be linked with the probable anticoagulation activity of TM; however, additional investigations in this field are needed.

To assess the therapeutic effect of the investigated semi-synthetic triterpenoids on colon inflammation, we performed immunohistochemical staining of colon sections with monoclonal antibodies for the pro-inflammatory cytokine tumor necrosis factor α (TNFα). Representative images displayed in Figure 5A demonstrate that a healthy colon is characterized by the basal expression of TNFα, which is in agreement with the published data [30]. DSS exposure significantly increased the TNFα level (Figure 5A, DSS, DSS+oil), whereas the treatment of DSS-induced colitis mice with SM and TM abrogated this effect. Both compounds decreased the expression of TNFα in the colon tissue up to the level of healthy mice, and their effectiveness was comparable to the reference drug SLZ (Figure 5A).

Finally, given the tight linkage between DSS-induced inflammation, oxidative stress and changes in the glutathione (GSH) metabolism [31] and, moreover, the ability of cyano enone-bearing triterpenoids to modulate the intracellular GSH level [11,32], we questioned whether TM and SM influenced the GSH level in DSS-injured colons. We showed that only TM increased the GSH production in colon tissue by 2.4-fold compared to the vehicle-treated group, whereas SM and SLZ did not influence GSH synthesis in this model (Figure 5B). Thus, the obtained data revealed that, along with its pronounced anti-inflammatory activity, TM also showed a high antioxidant effect in DSS-inflamed colon tissue.

### 2.4. Akt1, STAT3 and Dopamine Receptor D2 are Probable Primary Targets of TM, Associated with Its Anti-Colitic Activity

In order to understand the molecular mechanism of the anti-colitic effects of TM, we attempted to identify TM’s probable primary protein targets, associated with the development of acute DSS-induced colitis. To perform this, we first questioned which key genes could assemble the regulome of DSS-stimulated colitis in mice. The re-analysis of three independent microarray datasets from the Gene Expression Omnibus (GEO) repository (dataset identifiers: GSE64658, GSE42768, and GSE71920) using the GEO2R tool revealed three independent sets of differentially expressed genes (DEGs; fold change > 1.5, *p* < 0.05) in the acute phase of DSS-induced murine colitis in comparison with the healthy control. A subsequent Venn diagram analysis of the obtained DEGs revealed 188 DEGs, common to all the analyzed GEO datasets (colitis regulome) (Figure 6A).

Then, we predicted a list of the probable protein targets of TM using two independent ligand-based virtual screening platforms, including the polypharmacology browser 2.0 tool [33] and the STITCH database [34] (Appendix A). In order to reveal the probable targets of TM most related to the development of colitis, the protein–protein interaction (PPI) network was further reconstructed from the predicted TM’s protein targets and colitis regulome using the STRING database (Figure 6B). We ranked the probable targets of TM according to their level of interconnection within the PPI network and revealed the top five interconnected proteins (AKT serine/threonine kinase 1 (Akt1), tyrosine protein kinase Src, signal transducer and activator of transcription 3 (STAT3), dopamine receptor D2 (DRD2), and opioid receptor Mu 1 (OPRM1)), which were characterized to the highest degree and therefore, played an important role in the development of DSS-induced colitis.

To further validate the ability of TM to bind to the identified colitis-associated key nodes, we conducted molecular docking simulations. We found that the investigated triterpenoids could dock into the active sites of the revealed proteins, except for Mu opioid receptor OPRM1. As can be seen from Figure 6C, both compounds showed extremely high binding energy values for OPRM1, which indicated a low probability of interaction. A more detailed analysis of the docked poses of TM revealed that Akt1, Src, STAT3 and dopamine receptor D2 (Drd2) could be considered as the probable protein targets. TM forms a wide network of hydrophobic interactions and hydrogen bonds with the residues common for both TM and the well known inhibitors of these proteins. As shown in Figure 6C, the docking data suggested that the binding of TM to Akt1 involved a strong hydrogen bond with Glu234 (3.25 Å), playing a key role in the Akt1 biological activity [35].

The ability of TM to form a hydrogen bond with Asp404 (3.22 Å) of Src can confirm its probable anti-Src activity. Previously, comprehensive ligand-based pharmacophore modeling revealed that the compounds that showed a hydrogen bond interaction with Asp404 were considered as potent inhibitors of this kinase [36]. In the case of STAT3, TM formed five hydrogen bonds, including an interaction between the cyano group of ring A and Glu638 (3.09 Å), a key residue for a range of inhibitors hydrogen bonding to STAT3 [37] (Figure 6D). TM fit snugly into the binding site of the known DRD2 antagonist haloperidol, forming nonbonding contacts with the side chain of Asp114 (Figure 6D) and playing a key functional role in many G-protein coupled receptors [38].

The analysis of the docked poses of SM also confirmed its interactions with Akt1, Src, STAT3 and DRD2 (Figure 6D); however, in our opinion, only STAT3 can be considered as the probable pharmacology-relevant target. We showed the ability of SM to form a hydrogen bond with Arg609 (3.03 Å), which was crucial for the development of STAT3 inhibitors [39,40], whereas in the case of other proteins, SM did not interact with functionally important residues or formed significantly less hydrophobic bonds compared to TM; see the DRD2–SM complex as an example (Figure 6D). Thus, the performed comprehensive in silico analysis showed that TM can control the development of colitis by targeting Akt1, Src, STAT3 and DRD2; however, in order to elucidate this mechanism more precisely, a detailed biochemical or biophysical analysis is required.

Finally, in order to shed light on the low efficiency of TM to reduce the level of visible blood in stool (Figure 3C) along with its promising anti-colitic activity (Figure 3, Figure 4 and Figure 5), the ability of TM to bind to thrombin, being a target of glycyrrhizin [29], was tested using a molecular docking simulation. Our results showed that TM can snugly fit into the active site of thrombin in a position very close to that of bis(phenyl)methane, a known inhibitor of thrombin (Figure 7A). As shown in Figure 7B, the binding of TM to thrombin involves two strong hydrogen bonds between the carbonyl oxygen at C-30 and the carbonyl group at C-19 with the side chains of His57 (3.16 Å) and Gly216 (2.90 Å), respectively, being the component of a catalytic triad and a key residue for a series of inhibitors in their hydrogen bonding to thrombin [41].

The triterpenoid nucleus of TM was stabilized by a range of hydrophobic residues, including Tyr60A and Tyr60B, which are important for thrombin recognition from small synthetic inhibitors [42], and Ser214 and Glu217, playing a key role in the interaction of the peptide substrate with thrombin [43]. SM, due to the absence of an additional carbonyl group at C-19, formed only one hydrogen bond with His57 (3.13 Å) and its binding was characterized by significantly higher energy (∆G^SM^ = −6.5 kcal/mol) compared to TM ((∆G^TM^ = −7.3 kcal/mol). Thus, the molecular docking data were quite consistent with the presence of blood in the stool of the TM-treated DSS-exposed mice; however, more detailed investigations in this field are required.

The obtained results correlated well with the published data. Previously, research demonstrated that the PI3K/Akt, STAT3 and dopamine receptor D2 signaling pathways were involved in the pathogenesis of ulcerative colitis and inflammatory bowel disease and that their inhibition effectively ameliorated these pathologies [30,44,45,46]. Recently, Fitzpatrick et al. showed that CDDO-Im, a structural analog of TM and SM, effectively improved DSS-induced colitis in mice by inhibiting STAT3 activation in DSS-inflamed colon tissues [25]. According to Ahmad et al., bardoxolone methyl (Figure 1) could directly interact with STAT3, leading to the inhibition of its activity [47]. Conversely, the inhibition of Src aggravated the DSS-stimulated colitis, inducing a more severe inflammation response, accompanied by high TNFα and low IL-10 expression [48]. According to our immunohistochemical data, TM effectively inhibited the production of TNFα in colon tissue of DSS-exposed mice (Figure 4A). Therefore, the probable anti-Src activity of TM, revealed by molecular docking (Figure 6D), did not play a decisive role in the regulation of the anti-colitic effects of TM.

## 3. Materials and Methods

### 3.1. General Experimental Procedures

The melting point was determined on a FP900 thermosystem (Mettler Toledo, Greifensee, Switzerland) and were uncorrected. The elemental composition of the products was determined from high-resolution mass spectra recorded on a DFS (double focusing sector) Thermo Electron Corporation instrument (Thermo Electron Corporation, Langenselbold, Germany). The optical rotations were measured with a PolAAr 3005 polarimeter (Optical Activity Ltd., Cambridgeshire, UK). The IR spectra were recorded with Bruker Vector 22 spectrometer (in KBr pellets). ^1^H and ^13^C-NMR spectra were measured from CDCl_3_ solutions on Bruker spectrometers: DRX-500 (500.13 MHz for ^1^H and 125.76 MHz for ^13^C) and AV-400 (400.13 MHz for ^1^H and 100.61 MHz for ^13^C) (Bruker BioSpin GmbH, Rheinstetten, Germany). The chemical shifts were recorded in *δ* (ppm) using *δ* 7.24 (^1^H-NMR) and *δ* 76.90 (^13^C-NMR) of CHCl_3_ as internal standards. Chemical shift measurements were given in ppm and the coupling constants (*J*) in hertz (Hz). The structure of the compounds was determined by NMR using standard one-dimensional and two-dimensional procedures (^1^H-^1^H COSY, ^1^H-^13^C HMBC/HSQC, ^13^C-^1^H HETCOR/COLOC. The purity of the final compound for biological testing was >98% as determined by HPLC analysis. The HPLC analyses were carried out on a MilichromA-02, using a ProntoSIL 120-5-C18 AQ column (BISCHOFF, 2.0 × 75 mm column, grain size 5.0 μm). The mobile phase was Millipore purified water with 0.1% trifluoroacetic acid at a flow rate of 150 µL/min at 35 °C with UV detection at 210, 220, 240, 260, 280, 300, 320 and 360 nm. A typical run time was 25 min with a linear gradient of 0–100% methanol. The flash column chromatography was performed with silica gel (Merck, 60–200 mesh and neutral alumina (Chemapol, 40–250 mesh)). A reaction course was monitored by TLC analysis using Merck 60 F254 silica gel on aluminum sheets with the eluent CHCl_3_; CHCl_3_–AcOEt (25:3) and the plots were visualized using UV light.

Methyl 2-cyano-3-oxo-18βH-olean-1(2),12(13)-dien-30-oate was synthesized according to a described method [16], and selenium (IV) oxide was purchased from abcr GmbH & Co. KG; all the solvents were purified and dried according to standard procedures.

### 3.2. Synthesis of the Methyl 2-cyano-3,12,19-trioxo-olean-1(2),9(11),13(18)-trien-30-oate (Trioxolone methyl) (TM)

Selenium dioxide (10.58 g, 95.1 mmol, 8 equiv.) was added to a solution of methyl 2-cyano-3-oxo-18βH-olean-1(2),12(13)-dien-30-oate (5.84 g, 11.9 mmol, 1 equiv) in glacial AcOH (150 mL) in two equal portions with an interval of 3h. The reaction mixture was heating under reflux for about 6 h (the reaction course was monitored by TLC and ^1^H-NMR) then cooled to room temperature and filtered. The filtrate was diluted with an equal volume of water and then it was extracted with the mixture CH_2_Cl_2_-Et_2_O (1:2, *v*/*v*) (3 × 150 mL). The combined organic phase was washed with H_2_O, aq NaHCO_3_ and brine before being dried over anhydrous MgSO_4_. The solvent was removed to give a yellow amorphous solid (5.54 g). The crude product was purified by two successive flash column chromatographies: (alumina, CHCl_3_) and (silica gel, 10–50% (*v*/*v*) AcOEt in *n*-hexane). Trioxolone methyl (TM) (3.14 g, yield 51%) was obtained as a white amorphous solid (98 + % HPLC purity).

m.p. 239–240 °C. [α^25^_D_] -54 (*c* 0.25 g/100 mL; CHCl_3_). HRMS: *m*/*z* calc. for (C_32_H_39_NO_5_)^+^ 517.6568; found 517.2828. IR (KBr, ν·cm^−1^): 2952, 2925, 2871, 2857; 2235 (CN); 1737, 1691, 1658, 1614 (C=O); 1463, 1384, 1261, 1151, 1043, 758.

^1^H-NMR (CDCl_3_, 400 MHz): *δ* = 1.03 (s, 3H, CH_3_-28), 1.13 (s, 3H, CH_3_-27), 1.14 (s, 3H, CH_3_-24), 1.19 (s, 3H, CH_3_-23), 1.29 (dddd, 1H, *J_15e,15a_* 13.5, *J_15e,16a_* 3.8, *J_15e,16e_* 3.2, H-15e), 1.34 (s, 3H, CH_3_-26), 1.48 (s, 3H, CH_3_-25), 1.57 (s, 3H, CH_3_-29), 1.51-1.82 (m, 10H, H-5a, 2H-6, 2H-7, 2H-16, H-21e, 2H-22), 1.87 (ddd, 1H, *J* 13.5, *J_15a,16a_* 13.5, *J_15a,16e_* 4.0, H-15a), 2.58 (ddd, 1H, *J_21a,21e_* 13.7, *J_21a,22a_* 13.7, *J_21a,22e_* 5.3, H-21a), 3.73 (s, 3H, OCH_3_-31), 6.28 (s, 1H, H-11), 8.03 (s, 1H, H-1).

^13^C-NMR (CDCl_3_, 100 MHz): *δ* = 166.1 (d, C-1), 114.6 (s, C-2), 199.4 (s, C-3), 45.4 (c, C-4 *), 48.0 (d, C-5), 17.9 (t, C-6), 32.0 (t, C-7), 45.3 (s, C-8 *), 169.8 (s, C-9), 42.85 (s, C-10), 123.7 (d, C-11), 186.5 (s, C-12), 138.6 (s, C-13), 44.8 (s, C-14), 25.0 (t, C-15), 33.7 (t, C-16), 41.9 (c, C-17), 152.5 (s, C-18), 205.9 (s, C-19), 59.4 (s, C-20), 31.9 (t, C-21), 36.1 (t, C-22), 27.1 (q, C-23), 21.3 (q, C-24), 27.7 (q, C-25), 27.1 (q, C-26), 22.07 (q, C-27), 22.13 (q, C-28), 18.5 (q, C-29), 172.8 (s, C-30), 52.0 (q, C-31), 114.1 (s, C-32).

### 3.3. Cell Cultures and Semi-Synthetic Triterpenoids SM and TM

The murine J774 macrophage cell line was obtained from the Russian Cell Culture Collection (Institute of Cytology, RAS, St. Petersburg, Russia). The J774 cells were cultured in Dulbecco’s modified Eagle’s medium (DMEM) (Sigma-Aldrich Inc., St. Louis, MO, USA) supplemented with 10% (*v*/*v*) heat-inactivated fetal bovine serum (BioloT, Russia) and an antibiotic-antimycotic solution (100 U/mL penicillin, 100 µg/mL streptomycin, 0.25 µg/mL amphotericin) and incubated at 37 °C in a humidified 5% CO_2_-containing air atmosphere (hereafter referred to as standard conditions). Semi-synthetic triterpenoids were dissolved in DMSO (stock solutions: 10 mM) and stored at −20 °C prior to use.

### 3.4. Cell Viability analysis by MTT Assay

The cells were seeded in tetraplicate in 96-well plates at density of 10^5^ cells per well. The plates were incubated under standard conditions for 24 h and were then treated with the tested compounds (0.1 and 1 µM) for an additional 24 h. After that, the MTT solution (10 µL, 5 mg/mL) was added to the cells and the incubation was continued under standard conditions for an additional 2 h. Then, the MTT-containing medium was aspirated and the formazan crystals formed in living cells were solubilized with 100 µL of DMSO. The absorbance of each well was read at test and reference wavelengths of 570 and 620 nm, respectively, on a Multiscan RC plate reader (Thermo LabSystems, Helsinki, Finland).

### 3.5. Measurement of LPS-induced NO Production by Macrophages

The J774 cells were seeded in pentaplicate in 96-well plates at a density of 10^5^ cells per well. The plates were incubated under standard conditions for 24 h. The cells were treated with LPS of *Escherichia coli* (*E. coli*) (10 µg/mL, serotype 055:B5, Sigma-Aldrich Inc., USA) with or without triterpenoids (0.05–1 µM) for an additional 24 h. The production of NO was evaluated by the measurement of the nitrite concentration in a culture medium by using a Griess Reagent System (Promega, Madison, WI, USA), according to the manufacturer’s instructions. The absorbance was measured at 570 nm in a Multiscan RC plate reader (Thermo LabSystems, Helsinki, Finland).

### 3.6. Mice

Female 8–10-week-old C57Bl/6 mice were purchased from the Vivarium of the Institute of Chemical Biology and Fundamental Medicine SB RAS (Novosibirsk, Russia). The mice were kept in plastic cages (5 × 7 animals per cage) under normal daylight conditions. Water and food were provided ad libitum. All the animal procedures were carried out in accordance with the recommendations for the proper use and care of laboratory animals (ECC Directive 2010/63/EU). The experimental protocols were approved by the Committee on the Ethics of Animal Experiments with the Institute of Cytology and Genetics SB RAS (Novosibirsk, Russia) (ethical approval number 56 from 10 August 2019).

### 3.7. Induction of Acute Colitis

Experimental colitis was induced by administration of 2.5% (*w*/*v*) dextran sodium sulfate (DSS, product #42867, Sigma-Aldrich Inc., USA) in drinking water for 7 days followed by a 3 day recovery (Figure 3A). The investigated compounds SM and TM were administered to DSS-treated mice via gastric gavage at a 5 mg/kg dose diluted in DMSO: sesame oil (1:9) simultaneously with DSS and within 3 days after its withdrawal (10 days in total) (*n* = 7 per group). The DSS-exposed animals treated with the vehicle only (DMSO: sesame oil (1:9)) or without treatment were used as control groups (*n* = 7 per group). Sulfasalazine (SLZ) (100 mg/kg) administered in DMSO: sesame oil (1:9) was used as a reference drug with proved anti-inflammatory effect regarding inflammatory bowel diseases (*n* = 7) [49,50]. The healthy group was treated with drinking water alone (*n* = 7).

To assess the colitis induction, the mice were examined daily for body weight and stool blood. The disease activity index (DAI) was determined according to the stool blood score as follows: 0—normal colored stool, 1—reddish stool, 2—bloody stool.

The mice were sacrificed 24 h after the last administration of the compounds by cervical dislocation. The colons were carefully separated from the proximal rectum, and the colon length was measured between the ileo–cecal junction and the proximal rectum. After mechanical cleaning with saline buffer, the colon tissues were collected for subsequent histological analysis and GSH assessment.

### 3.8. Histopathological Analysis

For the histological study the specimens of the colons were fixed in 10% neutral-buffered formalin (BioVitrum, Moscow, Russia), dehydrated in ascending ethanols and xylols and embedded in HISTOMIX paraffin (BioVitrum, Russia). The paraffin sections (5 μm) were sliced on a Microm HM 355S microtome (Thermo Fisher Scientific, Waltham, MA, USA) and stained with hematoxylin and eosin.

The expression of the neutral mucins in colon tissue was determined using Periodic Acid-Schiff (PAS) staining. The tissue sections were deparaffinized, rehydrated, and PAS-stained with Schiff’s reagent (BioVitrum, Russia) according to the standard protocol and then counterstained with hematoxylin.

For the immunohistochemical study the colon sections (3–4 μm) were deparaffinized and rehydrated. Antigen retrieval was carried out after exposure in a microwave oven at 700 W. The samples were incubated with the anti-TNFα specific monoclonal antibodies (ab1793, Abcam, Cambridge, MA, USA) according to the manufacturer’s protocol. Then, the sections were incubated with secondary horseradish peroxidase (HPR)-conjugated antibodies (Spring Bioscience detection system, Spring Bioscience, USA), exposed to the 3,3’-Diaminobenzidine (DAB) substrate, and stained with Mayer’s hematoxylin.

All the images were examined and scanned using Axiostar Plus microscope equipped with Axiocam MRc5 digital camera (Zeiss, Oberkochen, Germany) at magnifications of ×100 and ×400.

The histological quantitation of the colitis severity was performed using a scoring system comprising three independent parameters: (1) the severity of inflammation: 0—none, 1—slight, 2—moderate, 3—severe; (2) extent of inflammation: 0—none, 1—mucosal, 2—mucosal and submucosal, 3—transmural; (3) crypt damage: 0—intact, 1—loss of one-third of crypts, 2—loss of two-third of crypts, 3—entire crypt damage with surface epithelium intact, 4—entire crypt damage and epithelium lost. The score of each parameter was multiplied and all the numbers were summed. The maximum possible score was 10.

### 3.9. Glutathione (GSH) Assay

The colon tissue samples were weighted and homogenized in 1 mL of PBS followed by the centrifugation of the extract and the collection of the supernatant. The GSH level in the supernatants was measured using a GSH-Glo^TM^ Glutathione Assay kit (Promega, USA) according to the manufacturer’s instructions. The luminescence was detected using a CLARIOstar plate reader (BMG Labtech, Ortenberg, Germany).

### 3.10. GEO Datasets Processing

The gene expression profiles of GSE42768 (5 acute colitis samples, 5 control samples), GSE64658 (3 acute colitis samples, 6 control samples) and GSE71920 (3 acute colitis samples, 3 control samples) were uploaded from the Gene Expression Omnibus (https://www.ncbi.nlm.nih.gov/geo). The identification of the differentially expressed genes (DEGs) between the colitic and the healthy colon tissue samples was performed by a GEO2R tool (https://www.ncbi.nlm.nih.gov/geo/geo2r/) [51]. The *p* < 0.05 and |Fold Change| > 1.5 were considered as the cutoff value. A Venn diagram analysis of the revealed DEGs was carried out using InteractiVenn online tool (http://www.interactivenn.net/) [52].

### 3.11. Prediction of Probable Primary Targets of TM

The probable primary protein targets of TM were predicted by the Polypharmacology browser 2.0 tool (http://ppb2.gdb.tools/), using the ECfp4 Naïve Bayes Machine Learning model produced on the fly with 2000 nearest neighbors from the Extended Connectivity fingerprint ECfp4 (NN(ECfp4)+NB(ECfp4)) [33] and STITCH database (http://stitch.embl.de/) [34] according to developer’s instructions.

### 3.12. PPI Network Reconstruction

The protein–protein interactions (PPI) between the predicted primary targets of TM and the colitis-associated DEGs (being common for all 3 GEO datasets) were identified based on the data deposited in the STRING database [53] with a confidence score > 0.7. The created PPI network was visualized by Cytoscape 3.6.1. To calculate the level of interconnection of TM’s probable targets within the network, node degree scores were computed using the NetworkAnalyzer plugin [54].

### 3.13. Molecular Docking

The docking of TM with thrombin (Protein Data Bank (PDB) ID: 2ZDA), Akt1 (PDB ID: 6CCY), Src (PDB ID: 4O2P), STAT3 (PDB ID: 6NJS), DRD2 (PDB ID: 6LUQ) and OPRM1 (PDB ID: 5C1M) were carried out using Autodock Vina [55]. The 3D structures of the listed proteins were uploaded from RCSB Protein Data Bank (https://www.rcsb.org/). The co-crystalled ligands were extracted from the uploaded PDB files and the polar hydrogens and Gasteiger charges were added into the protein structures using AutoDock Tools 1.5.7. The 2D structures of TM and SM were converted into 3D form and their geometries were optimized with the MMFF94 force field using Marvin Sketch 5.12 and Avogadro 1.2.0, respectively. All the rotatable bonds within the ligand were allowed to rotate freely. The used docking parameters are listed in Table 1.

The best molecular interactions were identified based on the binding orientation of the proteins’ key residues and their corresponding binding energy values. The results were imported and analyzed using Discovery Studio Visualizer 17.2.0. The 2D plots of the protein–ligand interactions were analyzed using LigPlot+ 1.4.5.

### 3.14. Statistical Analysis

The data are expressed as the mean ± SD. The statistical analysis was performed using the two-tailed unpaired t-test. *p*-values of less than 0.05 were considered statistically significant.

## 4. Conclusions

In summary, trioxolone methyl (TM), a novel 18βH-glycyrrhetinic acid derivative, containing a 2-cyano-3-oxo-1(2)-en moiety in ring A and a 12,19-dioxo-9(11),13(18)-dien moiety in rings C, D and E, was synthesized from methyl 2-cyano-3,12,19-trioxoolean-1(2),9(11),13(18)-trien-30-oate by selenium oxide (IV) oxidation. We performed an analysis of the bioactivity of TM and revealed its high anti-inflammatory potential, which was more pronounced than that of SM. The formation of additional enone functionality in rings D/E, along with the enone system in ring C of SM, significantly reinforced the ability of the molecule to suppress NO production by LPS-activated J774 macrophages in vitro and effectively attenuated DSS-induced colitis in mice by inhibiting the DSS-stimulated damage and inflammatory infiltration of colon tissue. This was achieved through the suppression of neutral mucin and pro-inflammatory TNFα hyperproduction and the increase in the GSH level in the inflamed colon. The subsequent comprehensive in silico analysis revealed the ability of TM to bind to the active sites of key colitis-associated regulators, including Akt1, Src, STAT3 and dopamine receptor D2. Taken together, our results provided new insights into the potential mechanisms underlying the anti-inflammatory activity of cyano enone-bearing triterpenoids and revealed compound TM as a promising anti-inflammatory candidate.

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
