# Peer review of "Trioxolone Methyl, a Novel Cyano Enone-Bearing 18βH-Glycyrrhetinic Acid Derivative, Ameliorates Dextran Sulphate Sodium-Induced Colitis in Mice"

_molecules, 2020, doi:10.3390/molecules25102406_

Round 1
Reviewer 1 Report
This work, dealing with a novel cyano enone-bearing triterpenoid acid derivative, and of its activity against dextran sulphate sodium-induced colitis in mice, may be interesting for those who work in this field. I think it can be published in Molecules, with some major modifications.
Main concerns:
The Authors don’t report the statistic description of the single experiments, neither in the Figures, nor in the experimental section. So it is not clear how the experiments were carried out.
Only in the experimental section you can find the paragraph statistical analysis: “Data are expressed as the mean ± SD. Statistical analysis was performed using the two-tailed unpaired t-test. P-values of less than 0.05 were defined as statistically significant”. But is not specified the mean of what? 2,3,……or n experiments?
How much times were performed? duplicate, triplicate…..? This has to be specified.
For example a phrase like this “Results are expressed as the means ± SD of ….. experiments performed in ……..cate"
Minor concerns:
- There are too many acronyms in the abstract. Authors have to explain the meaning in the abstract, and the first time that the acronym appears along the text.
- The Authors in the abstract and in the conclusions affirm that “Altogether, our findings provide a basis for better understanding of the structure-activity relationship of cyano enone-bearing triterpenoids”. I think that is not possible to define SAR this study, because it is referred to only one derivative. Please, change this phrase.
- Page 1, line 38: change “consider” with “considered”.
- Page 2, line 46: I think some grammar mistake there is in the phrase: the formation an”. In any case, all this paragraph (line 47-54) should be re-written, because it is not clear. The Authors cite a C9 and a ring C, but the structure is shown after. They should better describe the structure and the numeration, with respect to the Figure 1, and improve Figure 1.
- Page 2, line 56. Why SM is defined bardoxolone methyl analog? From the structure it is a position isomer (the methyl ester moiety is in different positions of ring E).
- Page 3, scheme 1. The numeration of TM seems different from that of Figure 1.
- Page 3, line 99: delete “of” before compounds.
- Page 4, Figure 2B. What does “p = 2.6E-03” mean?
- Page 6, line 193: What is the meaning of “PAS”? It is explained in the experimental section but this is the first time that it appears.
- Page 7, Figure 4B: What does “p = 3.7E-05” mean?
- Page 7, lines 205-218. I think that these docking studies on thrombin inhibitor are only speculative. However, it should be better that the results are discussed together with the other docking studies.
- Page 8, line 258, 259: another time, acronyms without meaning “GSE64658, GSE42768 and GSE71920 using the GEO2R”.
- Page 8, line 260: venn needs the Capital letter.
- Page 10, line 291: “D1 (Drd2)”. What dopamine receptor refer to? D1 or D2? It is not clear. Please check thoroughly along the text, and report it in the same way (at line 301 you find 2DRD2 and in Figure 6 DRD2!)
- Page 10, line 302: “hydrophobic interaction with Asp114” It seems a little strange that aspartic acid does hydrophobic interactions
- Page 10, line 326: change “expreimental” with “experimental”
- Page 11, line 354: change “NMR 1H” with “1H NMR”
- Since TM is a solid, why the melting point is not described?
Page 14, line 498: another time “dopamine receptor D1”. Please correct (see page 10, line 291).
Author Response
Dear Reviewer #1,
We are sincerely thankful to you for your deep analysis of our manuscript and highly valuable remarks. We revised the manuscript according to your comments and, please, let us respond to your questions.
- 1. The Authors don’t report the statistic description of the single experiments, neither in the Figures, nor in the experimental section. So it is not clear how the experiments were carried out.
Only in the experimental section you can find the paragraph statistical analysis: “Data are expressed as the mean ± SD. Statistical analysis was performed using the two-tailed unpaired t-test. P-values of less than 0.05 were defined as statistically significant”. But is not specified the mean of what? 2,3,……or n experiments?
How much times were performed? duplicate, triplicate…..? This has to be specified.
For example a phrase like this “Results are expressed as the means ± SD of ….. experiments performed in ……..cate"
Authors: Corrected. It is our disappointing flaw. The measurements of macrophage viability and NO production were repeated by us three and two times in tetra- and pentaplicate, respectively. This phrase was introduced to the description of Figure 2 (please, see lines 135-136; marked by yellow). Besides this, the total number of mice per group (n = 7) was added to all Figures, containing in vivo data, and the section 4.7 in Materials and Methods (please, see lines 177, 203-204, 245, 453-457; marked by yellow).
- There are too many acronyms in the abstract. Authors have to explain the meaning in the abstract, and the first time that the acronym appears along the text.
Authors: Corrected. Please, see lines 19, 22, 25, 62, 64, 77, 82-83, 116-117, 140, 176, 205, 208, 225, 233, 252, 263-265, 269-270, 272-273, 274 (marked by yellow).
- The Authors in the abstract and in the conclusions affirm that “Altogether, our findings provide a basis for better understanding of the structure-activity relationship of cyano enone-bearing triterpenoids”. I think that is not possible to define SAR this study, because it is referred to only one derivative. Please, change this phrase.
Authors: Corrected. The final phrase of the abstract was rewritten (please, see lines 30-31; marked by yellow).
- Page 1, line 38: change “consider” with “considered”.
Authors: Corrected. Please, see line 39 (marked by yellow).
- Page 2, line 46: I think some grammar mistake there is in the phrase: the formation an”. In any case, all this paragraph (line 47-54) should be re-written, because it is not clear. The Authors cite a C9 and a ring C, but the structure is shown after. They should better describe the structure and the numeration, with respect to the Figure 1, and improve Figure 1.
Authors: Corrected. The current manuscript was undergone the English editing by MDPI English editing service (please, find attached certificate). In order to make the mentioned paragraph more clear, some modifications were introduced to the text (please, see lines 49-50) and, moreover, the structures of bioactive semi-synthetic triterpenoids, bearing cyano enone pharmacophore without Michael acceptor center at C-9, were introduced to Figure 1 (please, see line 47 and Figure 1).
- Page 2, line 56. Why SM is defined bardoxolone methyl analog? From the structure it is a position isomer (the methyl ester moiety is in different positions of ring E).
Authors: Corrected. Please, see line 59 (marked by yellow).
- Page 3, scheme 1. The numeration of TM seems different from that of Figure 1.
Authors: Corrected. The numeration of TM was re-checked and corrected.
- Page 3, line 99: delete “of” before compounds.
Authors: Corrected.
- Page 4, Figure 2B. What does “p = 2.6E-03” mean?
Authors: Corrected. This type of p-value representation is often used in transcriptomic studies. p = 2.6E-03 means p = 0.0026. We rewrote p-value in conventional form.
- Page 6, line 193: What is the meaning of “PAS”? It is explained in the experimental section but this is the first time that it appears.
Authors: Corrected (please, see line 208).
- 11. Page 7, Figure 4B: What does “p = 3.7E-05” mean?
Authors: Corrected. Please, see Figure 4B.
- Page 7, lines 205-218. I think that these docking studies on thrombin inhibitor are only speculative. However, it should be better that the results are discussed together with the other docking studies.
Authors: Corrected. The paragraph, describing the docking studies on thrombin, was transferred to section 2.3 (Please, see lines 313-329, Figure 7).
- Page 8, line 258, 259: another time, acronyms without meaning “GSE64658, GSE42768 and GSE71920 using the GEO2R”.
Authors: Corrected. Please, see line 252.
- Page 8, line 260: venn needs the Capital letter.
Authors: Corrected. Please, see line 255.
- Page 10, line 291: “D1 (Drd2)”. What dopamine receptor refer to? D1 or D2? It is not clear. Please check thoroughly along the text, and report it in the same way (at line 301 you find 2DRD2 and in Figure 6 DRD2!).
Authors: Corrected. We are very grateful to you for your careful analysis of the manuscript. It is our disappointing misprint! TM can target dopamine receptor D2. All misprints were corrected (please, see lines 291, 362; marked by yellow).
- Page 10, line 302: “hydrophobic interaction with Asp114” It seems a little strange that aspartic acid does hydrophobic interactions
Authors: Corrected. It was our inaccuracy. According to LigPlot+ official site, the combs on the 2D docking representation figures represent nonbonding contacts, not hydrophobic interactions. The corrections were introduced into the text (please, see lines 283-284, 302, 334).
- Page 10, line 326: change “expreimental” with “experimental”
Authors: Corrected.
- Page 11, line 354: change “NMR 1H” with “1H NMR”
Authors: Corrected.
- Since TM is a solid, why the melting point is not described?
Authors: Corrected. The melting point was introduced to section 4.2 (please, see line 403 and lines 367-368 (the information about used instrument for melting point determination).
- Page 14, line 498: another time “dopamine receptor D1”. Please correct (see page 10, line 291).
Authors: Corrected.

Reviewer 2 Report
This paper is well written and there is a logical and thorough analysis of the experimental results. This paper incorporates a good combination of chemistry, various biological methods and molecular modelling. The authors have mentioned that bardoxolone methyl reached phase III clinical trial, could this be used as a reference drug in addition to sulfalazine?
There are minor but recurrent grammatical issues with parts of this paper that need to be addressed before it can be published. The author who wrote the biology component of section 2.2 has an excellent command of written English, it would be advisable for them to make the appropriate changes to the rest of the paper.
In addition to the grammatical changes I would suggest that the following changes be made to the manuscript.
Upfield and downfield shifts should be used instead of strong and weak fields to describe the shift in NMR resonances.
Line 101 change: kept their place at ~1.14 ppm and ~1.47 ppm, respectively
Line 368: The signals between 1.51-1.82 ppm need to be included even if they are multiplets which cannot be analysed further. If there are continuous NMR signals between 1.51 and 1.82 ppm, then 1.51-1.82 (m, xH) (x is the number of hydrogens) is fine.
Figure 4E and Figure 6D: I cannot read the atom numbers in either diagram, the font size of the numbers needs to be increased and perhaps restrict the numbering to atoms which interact with amino acid residues.
The paper would read better if the conclusion was directly after the results section, rather than at the end of the experimental methods section.
Author Response
Dear Reviewer #2,
Thank you for careful study of our manuscript and for your very useful remarks and comments. We revised the manuscript and, please, let us respond to your questions and comments.
- The authors have mentioned that bardoxolone methyl reached phase III clinical trial, could this be used as a reference drug in addition to sulfalazine?
Authors: In the current study we used sulfasalazine as a reference agent since it is a key drug for treating ulcerative colitis. Bardoxolone methyl due to its marked anti-inflammatory and cytoprotective activity can be also considered as a reference drug in this model, especially for investigations of bioactivities of cyano-enone-bearing compounds. Thank you for your advice! We will use bardoxolone methyl as a reference agent in our next studies.
- There are minor but recurrent grammatical issues with parts of this paper that need to be addressed before it can be published.
Authors: Corrected. The current manuscript have been processed by MPDI English Editing Service (please, find attached certificate, confirming this).
- Upfield and downfield shifts should be used instead of strong and weak fields to describe the shift in NMR resonances.
Authors: Corrected. Please, see lines 102, 104, 106, 111 (marked by yellow).
- Line 101 change: kept their place at ~1.14 ppm and ~1.47 ppm, respectively
Authors: Corrected. Please, see line 103.
- Line 368: The signals between 1.51-1.82 ppm need to be included even if they are multiplets which cannot be analysed further. If there are continuous NMR signals between 1.51 and 1.82 ppm, then 1.51-1.82 (m, xH) (x is the number of hydrogens) is fine
Authors: Corrected. Please, see line 408.
- Figure 4E and Figure 6D: I cannot read the atom numbers in either diagram, the font size of the numbers needs to be increased and perhaps restrict the numbering to atoms which interact with amino acid residues.
Authors: Corrected. The numbers of atom which interact with amino acid residues were increased (Please, see Figure 6 and Figure 7).
- The paper would read better if the conclusion was directly after the results section, rather than at the end of the experimental methods section.
Authors: Corrected. The chapter “Conclusions” was introduced directly after the section, describing the results (Please, see lines 349-364).

Round 2
Reviewer 1 Report
The Authors generally revised the manuscript following my comments.
Some mistakes and misprints are still present:
Line 136: terta- (A) instead of tetra
Line 362: I said ”The Authors in the abstract and in the conclusions affirm that “Altogether, our findings provide a basis for better understanding of the structure-activity relationship of cyano enone-bearing triterpenoids”. I think that is not possible to define SAR this study, because it is referred to only one derivative. Please, change this phrase”. The Authors changed this phrase only in the abstract. Also the conclusions have to be change
Figure 2B: Authors: Corrected. This type of p-value representation is often used in transcriptomic studies. p = 2.6E-03 means p = 0.0026. We rewrote p-value in conventional form. It is not true. p = 2.6E-03 is present also in this revised form
Author Response
Dear Reviewer #1,
We are deeply grateful to you for your careful analysis of the corrected version of the manuscript and sincerely apologize for the revealed annoying mistakes. We revised the manuscript according to your comments and corrected all misprints and omissions.
- Line 136: terta- (A) instead of tetra
Authors: Corrected (please, see line 136, marked by yellow).
- Line 362: I said ”The Authors in the abstract and in the conclusions affirm that “Altogether, our findings provide a basis for better understanding of the structure-activity relationship of cyano enone-bearing triterpenoids”. I think that is not possible to define SAR this study, because it is referred to only one derivative. Please, change this phrase”. The Authors changed this phrase only in the abstract. Also the conclusions have to be change.
Authors: Corrected. The last sentence into the Conclusions section was rewritten (please, see line 361-362, marked by yellow).
- Figure 2B: Authors: Corrected. This type of p-value representation is often used in transcriptomic studies. p = 2.6E-03 means p = 0.0026. We rewrote p-value in conventional form. It is not true. p = 2.6E-03 is present also in this revised form.
Authors: Corrected (please, see Figure 2; marked by yellow).